# Influence of Cutting-Edge Microgeometry on Cutting Forces in High-Speed Milling of 7075 Aluminum Alloy

**DOI:** 10.3390/ma16103859

**Published:** 2023-05-20

**Authors:** Łukasz Żyłka, Rafał Flejszar, Paweł Lajmert

**Affiliations:** 1Department of Manufacturing Techniques and Automation, The Faculty of Mechanical Engineering and Aeronautics, Rzeszow University of Technology, W. Pola 2 Str, 35-959 Rzeszow, Poland; r.flejszar@prz.edu.pl; 2Faculty of Mechanical Engineering, Lodz University of Technology, Stefanowskiego 1/15 Str, 90-537 Lodz, Poland; pawel.lajmert@p.lodz.pl

**Keywords:** milling, cutting-edge microgeometry, cutting forces

## Abstract

In the present study, the impact of cutting-edge microgeometry on the cutting forces in the finish milling of a 7075-aluminium alloy was analysed. The influence of selected values of the rounding radius of cutting edge, and the size of the margin width, on the cutting-force parameters was analysed. Experimental tests were carried out for different cross-sectional values of the cutting layer, changing the feed per tooth and radial infeed parameters. An analysis of the various statistical parameters of the force signal was performed. Experimental mathematical models of the relationship of the force parameters to the radius of the rounded cutting edge and the width of the margin were developed. The cutting forces were found to be most strongly influenced by the width of the margin and, to a minor extent, by the rounding radius of the cutting edge. It was proved that the effect of margin width is linear, and the effect of radius R is nonlinear and nonmonotonic. The minimum cutting force was shown to be for the radius of rounded cutting edge of about 15–20 micrometres. The proposed model is the basis for further work on innovative cutter geometries for aluminium-finishing milling.

## 1. Introduction

In recent years, in the field of manufacturing engineering, researchers have increasingly described unusual tools [1,2] and high-tech manufacturing systems [3], focused on the development of new machining methods [4] and strategies [5]. However, even then, research into the cutting process still very often turns to the topics of forces, stability, surface quality, or productivity. As an example, see the work of Grabowski et al. [6], in which they considered the prediction of cutting forces and process stability in serrated-end milli cutters. When the simulation and experimental data of both serrated and variable helix cutters were compared, it became apparent that process stability and cutting forces could be predicted quite accurately. They also showed that the ways in which milling cutter geometries are modified have a significant effect on the stability of the process itself. This is the conclusion also reached by Burek et al. [7,8]. They investigated different types of serrated milling cutters, based on which they concluded that the shape of the cutting edge plays an important role in both the effect on forces and vibrations in the high-performance rough-milling process of 7075-aluminium alloy. A similar issue was raised by Koca and Budak [9]; however, they focused on optimising the shape of serrations to reduce cutting forces. The optimum geometry was found to allow for much larger ranges of permissible cutting parameters and a significant increase in the areas of stability. New cutting-force models are also frequently developed, for example, the multiparameter milling force model for C45 steel proposed by Kolar et al. [10]. They carried out cutting tests for different rake, clearance and helix angles at variable cutting speeds, feeds per tooth, and depths of cut, taking into consideration tool wear. A total of 270 differentiated trials were performed, from which a mathematical model was developed using linear regression.

Various ways of simulating cutting processes are used very frequently instead of empirical mathematical models [11,12]. One aspect addressed by researchers is the simulations performed to optimise cutter geometry. An example is the work [13] of Li et al., in which they used the three-dimensional finite element method to design and optimise the cutter geometry. They used a reduction in the cutting force and process temperature as determinants of the appropriate process. The selection of the right angle of the helix and the shape of the groove (taking into account the angle of the blade and the number of blades) was an optimal choice. This research can be summarised by the fact that the simulation succeeded in developing the optimum cutter geometry for the machining of the difficult-to-machine Ti-6Al-4V alloy.

The cutting process itself is significantly influenced not only by technological parameters and tool geometry, but also by the microgeometry of the cutting edge. In their 2014 article [14], Denkena and Biermann conducted a comprehensive review of the literature on the topic of manufacturing technology and the influence of cutting-edge microgeometry on the cutting process. They showed great interest in research on the topic of cutting-tool microgeometry. However, they concluded that many aspects still need to be considered fully understand its influence on the cutting process. However, more sources indicate an increased interest in this issue. In their 2022 review paper [15], Li et al. took on the task of compiling knowledge on cutting performance and surface integrity under the influence of cutting-tool microgeometry. It is clear from the text that researchers are mostly concerned with the problems of the turning process. In the case of milling, the number of published papers is much lower. One of the articles that addresses this issue is the work of Fulemova and Janda [16] that examined how the radius of a cutting insert and the way it was made affect tool life, surface roughness, and forces in the cutting process. They worked with ferrite–martensite stainless steel and noticed that an increase in the radius of the rounded cutting insert increases tool life. It turned out that better results were obtained for the tool made by drag-finishing technology than for the grinded and laser-treated tools. The effect of the microgeometry of tools made by this technology was also presented in the work of Lv et al. [17]. They conducted an austenitic stainless-steel experiment and observed that the cutting force increased as the radius value of the circle increased, although the useful life of the tool was still increasing. They showed that, using specific cutting parameters, the radius of the rounded blade can result in an increase in forces of up to approximately 125%. They also observed that the number of defects on the tool surface is closely related to the radius of the rounding, and the technology used to round it caused the number of defects present on the tool surface to decrease. Not only the value of the radius itself, but also its shape, can affect the process. Padmakumar and Shiva Pradeep [18] investigated the effect of the form factor in the milling of chromium–molybdenum alloy steel. This factor illustrates how symmetrical the roundness of the cutting edge is. When comparing symmetrical rounding to one of the nonsymmetrical types, it was found that, using specific cutting parameters, approximately 30% higher total force and more than 50% worse surface roughness could be observed with symmetrical rounding as opposed to inverted waterfall rounding. The biggest disadvantage of the tested geometry is that the roundness coefficient decreases as the machining process proceeds, which means that the geometry will only work well for short processes. The question of the influence of microgeometry, or more precisely, the radius of the rounded cutting edge, on roughness was addressed by Hronek and Zetek in their work [19]. In the down-milling of the Inconel 718 alloy with milling cutters with different roundings of the cutting edge, they noted that the roughness of the machined surface increases with increasing radius of the rounding. However, the study showed that this is not a linear relationship and that for some changes in radius values, no significant change in roughness can be seen. A study [20] of the high-carbon steel cutting process by Zhao et al. can serve as a confirmation. They proved that there is a critical radius of roundness of the cutting edge, from which much better roughness results can be obtained.

Over the years, tool manufacturers have been trying to design tools that, despite high cutting parameters, will still maintain low surface roughness and ensure a stable machining process. Wiper geometry is one of these types of solutions. In a nutshell, it is a surface on the cutting insert that smooths the machined surface. However, this often results in increased friction between the workpiece and the tool, which can cause increased forces and instabilities in the process. Research on this type of geometry was done by Padmakumar and Sathish Kumar [21] on the face milling of high-carbon steel. They tested how forces and surface quality would behave if the microgeometry of the cutting insert were changed. They tested both sharp, chamfered, and rounded cutting edges, as well as those prepared with wiper technology. The ANOVA analysis they performed showed that microgeometry was the most significant factor influencing surface roughness, followed by cutting speed and feed rate. Continuing this research, Padmakumar [22] addressed optimisation of wiper geometry during face milling of chrome–molybdenum steel. He compared geometries with just a rounding and one in which the section between the rounding of the cutting insert and the cutting face is flat. Similarly, macrogeometries were analysed by Toledo et al. [23] in medium-tensile low-hardenability carbon-steel face milling. As a result of all the aforementioned studies, it can be concluded that choosing the right wiper geometry significantly reduces roughness, increases productivity, and ensures longer tool life.

A comprehensive review of the literature did not identify any published papers that dealt with the topic of the microgeometry of the cutter’s margin, the cylindrical surface located between the cutter’s rake and the clearance surface (similar to reamers). The idea is a combination of two aspects, the margin found in drills or reamers and wiper technology. The idea is that the introduction of a cylindrical surface should smooth the surface, but increased friction can create instability in the process. The principal objective of this study was to analyse the influence of selected microgeometry parameters of a milling cutter on the machining of 7075-aluminium alloy on cutting forces and to develop empirical mathematical models of the cutting forces.

## 2. Materials and Methods

### 2.1. Materials and Tools

The 7075-aluminium alloy, which is one of the most frequently used alloys in the aerospace industry, was applied as the material for the study. The test specimens were cut from 100 mm × 100 mm square bars using a band saw. To mount the samples on the piezoelectric dynamometer, holes and countersinks were prepared in the samples for the mounting screws.

The tools used during the experiments were made by Engram. They were 16 mm diameter carbide monolithic milling cutters, with three blades, an overall length of 90 mm, a working section of 40 mm, a 45-degree helix angle, a rake angle of 12 degrees and a second clearance angle of 25 degrees. Each of the cutters was characterised by varying microgeometry of the margin, and the radius of the cutting-edge rounding (Figure 1).

The microgeometry was made using abrasive jet-machining technology and the tools were coated with 2-micrometer-thick ZrN coating using PVD technology, which is dedicated to the machining of non-ferrous materials in order to minimise friction. Due to the manufacturing tolerances of the individual cutter dimensions, the width of the margin and the rounding radii were measured before the test for a more accurate analysis. An Alicona InfiniteFocus optical microscope was used to measure the width of the margin and the radius of rounding of the cutting edge. The measurements results were collected for the subsequent development of appropriate models in Table 1.

The tests were performed using a DMU 100 Monoblock machine, manufactured by DMG Mori. A KISLER 9257B multicomponent piezoelectric dynamometer with maximum forces up to 10 kN was mounted on the machine table (Figure 2). A test sample was fixed to it using four clamping screws. The measured signal was transmitted to a Kistler 5070 charge amplifier, next to a digital analogue converter NI USB-6003 from National Instruments, and then to a computer with NI SignalExpress software. The signal was scaled directly in the software to values in Newton. The sampling frequency was assumed to be 12 kHz, but the amplifier cut off all frequencies above 2 kHz.

Cutting tests were carried out by side milling with a constant axial depth of cut. The cutting-force components were measured in the X and Y directions, in the feed direction and in the direction normal to the feed (Figure 3).

The following constant cutting parameters were adopted: cutting speed *v_c_* = 900 m/min, rotational speed *n* = 18,000 rpm and axial depth of cut *a_p_* = 20 mm. The variable parameters during the cutting tests were: feed per tooth *f_z_* = 0.06; 0.08; 0.1 mm/tooth and radial depth of cut *a_e_* = 0.4; 0.7; 1.0 mm. The experimental tests were carried out according to the statistical plan, central composite fractional design with three levels and two factors according to Table 2. External cooling was applied during all cutting tests with the use of Fuchs Ecocool 68 CF semisynthetic emulsion with a concentration of 6%.

Eight machining passes were carried out on one test piece. It was assumed that the rigidity of the rotary table drive is much greater than the rigidity of the individual axes, so the test piece was rotated on the rotary table each time it was machined (the machining pass was always made on the same axis). This ensured that the rigidity of the system was always constant. As the test piece was machined on two levels, it was always pre-machined before the actual machining pass, to ensure that the machining allowance was always the same.

### 2.2. Force Signal Parameters

During the tests, the force components were measured in the *X* and *Y* directions. For the analysis of the cutting process, the resultant force of these components was assumed, designated in the next part of the work as *F_XY_*. This resulting force was calculated as the root of the sum of the squares of the forces component *F_X_* and *F_Y_*:(1)$$F_{XY}=\sqrt{F_{x}^{2}+F_{y}^{2}}$$

An exemplified course of the resultant force component *F_XY_* obtained for *a_e_* = 0.7 mm, *f_z_* = 0.1 mm/tooth, *R* = 25.1 µm and *MAR* = 44.9 µm is shown in Figure 4.

From the recorded course of the resultant force component *F_XY_*, a fragment of about 0.4 s width was taken for the final cutting process analysis (see Figure 4). Additionally, before cutting tests, a modal analysis of the most flexible system, i.e., the tool-holder system, was carried out. The spectral characteristic of the system response for impulse excitation is shown in Figure 5.

The dominant frequency component of about 1800 Hz is well visible. Furthermore, the frequency of force changes corresponding to successive material removal by cutter flutes was 900 Hz (three flutes, *n* = 18,000 rpm). Therefore, to remove dynamic force components, corresponding to vibrations of the tool-holder system, a low-pass filter with a cutoff frequency of 1150 Hz was applied. Then, to characterise the cutting process, the mean force value *MeanF* and the mean value of the upper force envelope composed of the local force peaks value *MeanUpF*, corresponding to the successive cutter teeth, were calculated. In addition, the maximum *MaxF*, minimum *MinF* and dispersion *DispF* values of the resultant cutting force were determined. The course of the selected range of the resulting cutting force *F_XY_* from Figure 4 together with the descriptive parameters mentioned is shown in Figure 6.

In addition, for each force course, the standard deviation *StdDevF*, sample entropy *SampEntrF*, and approximate entropy *AppEntrF* from the course of the resultant cutting force were determined. These parameters are frequently used in the literature [24,25,26] to determine repeatability, predictability, and the degree of stationarity of the cutting process, which affects the final state of macrogeometry of the workpiece surface. The standard deviation describes how widely the signal values are spread around its mean:(2)$$StdDevF=\sqrt{\frac{1}{n-1}\overset{n}{\underset{i=1}{\sum }}{\left(F_{i}-\bar{F}\right)}^{2}},$$
where *n*—is a number of elements in the recorded time series of the force signal *F(i)* and $\bar{F}$ is the average of this time series.

Other interesting measures of time-series complexity and stationarity are the approximate entropy *AppEntr* [27] and the sample entropy *SampEntr* [28]. The sample entropy is an improved version of the approximate entropy because it does not include self-similar data patterns. Given the dataset in the form of time-series $X_{N}=\left\{x_{1},\;x_{2},\;x_{3},\ldots ,\;x_{N}\right\}$ we define data patterns of length *m*:(3)$$X_{m}\left(i\right)=\left\{x_{i},\;\;x_{i+1},\ldots ,\;x_{i+m-1}\right\}\;\;\;\;\;\;\left(1\leq i\leq N-m+1\right)$$
and the distance function *d* being the Chebyshev distance:(4)$$d\left[X_{m}\left(i\right),X_{m}\left(j\right)\right]\;\;\;\;\;\left(i\neq j\right)$$

The sample entropy can be defined as:(5)$$SampEntr=-log\frac{A}{B}$$
where *A* is the number of vectors pairs for which $d\left[X_{m+1}\left(i\right),\;\;X_{m+1}\left(j\right)\right]<err$, *B* is the number of vectors pairs for which $d\left[X_{m}\left(i\right),\;X_{m}\left(j\right)\right]<err$, *err* is a parameter selected on the basis of the standard deviation of the time series.

The *err* parameter can be selected independently for each analysed time series and depends on the standard deviation of the analysed time-series. Usually, the *err* parameter is assumed to be equal to 0.2 times the standard deviation of the signal, and the length of the data subsequence *m* equals 2, which most often gives satisfactory results [25]. A time series that contains many repetitive patterns has a relatively small sample entropy, while a less predictable process has a higher sample entropy [26].

## 3. Results and Discussion

### 3.1. Cutting-Force Parameters Analysis

Technological tests were performed for 12 tools. For each tool, 13 cutting tests were performed. In each test, the components of the cutting force were measured, and the value of the resulting force *F_xy_* was determined. A total of 156 cutting tests were carried out. Then, the values of the statistical parameters of the resulting cutting forces were determined. The results are shown in the graphs presented in Figure 6. Individual points over the tool number represent values obtained in nine tests carried out with different cutting parameters.

The basic parameter of the cutting force is its mean value, as shown in Figure 7a. It can be clearly seen that the lowest values of the *MeanF* parameter were obtained for tools No. 4 and 8. The highest values were registered for tools No. 5, 9 and 10, which are the tools with the highest *MAR* or *R* values. These values also have the largest dispersion. This shows that the mean milling force depends on the parameters of the tool microgeometry studied.

The milling process is a periodically variable process in which the milling cutter edges periodically plunge into the material. This results in a variable cross-section of the cut layer and, at the same time, a periodically variable cutting force. Therefore, the value of the cutting force can be determined in various ways. The most commonly used parameter is the average value of the cutting force. However, in reality, the tool and the workpiece are affected by the maximum force rather than the average force. Therefore, a very good force parameter in the milling process is the mean value of the upper envelope. It means the averaged maximum value of the cutting force from one pass. Analysing the values of the *MeanUpF* parameter, it can be seen that the lowest and most concentrated values were obtained for tools No. 4 and 8 (Figure 7b). For the other tools, the values of the mean maximum force are even twice as high. The determined values of the average of the upper envelope of the signal are characterised by higher values than the average force, but the distribution of the values of both parameters is almost identical. Therefore, the *MeanUpF* parameter should be considered to be the most suitable for analysing the milling process.

The maximum and minimum values of the cutting force registered in each cutting test were also analysed, as well as the dispersion between them *DispF* (Figure 7c–e). The *MinF* and *MaxF* values have a distribution similar to the *MeanF* cutting force. It can be clearly seen that the lowest values of the maximum and minimum cutting force occurred for tools 4 and 8. In contrast, the highest values of *MinF* and *MaxF* occurred for tools 1, 5, 9 and 10. From this it can be concluded that the use of the smallest margin width for a radius *R* of about 10 µm and 20 µm results in the lowest cutting forces. The use of the smallest margin width for a larger radius *R* results in the occurrence of much higher cutting forces and a lower stability and repeatability of the cutting process. The *DispF* parameter can be a measure of the stability of the process. The smaller the difference between the maximum and minimum value of the cutting force, the smaller the dynamics of the process. Figure 7c shows that only tools No. 4 and 8 provide the greatest stability of the cutting process.

The standard deviation illustrates the dispersion of force values around the mean force value. Therefore, it can be assumed to be a measure of the stability and repeatability of the cutting process. Analysis of the graph in Figure 7f clearly proves that the lowest values of *StdDevF* were obtained for tools No. 3, 4 and 8, which have very small values of the radius *R* or margin *MAR*. In contrast, the highest *StdDevF* values were obtained for tools 1, 2, 5, 6, 9 and 10, which are the tools with the largest margin width values above 76 micrometers. Hence, it can be concluded that the larger the margin width, the greater the dispersion of the cutting force, and the machining process is less predictable and stable.

Further parameters that can be taken as a measure of the stability and repeatability of the cutting process are the entropy parameters *AppEntr* and *SampEntr*. The entropy value of a disordered system was assumed to be greater than that of an ordered system. Analysing the graphs in Figure 7g,h, it can be observed that the lowest and most concentrated entropy was obtained for tools 4, 8, and 9. For these tools, the dispersion of *AppEntr* and *SampEntr* values was the smallest and the values of these parameters were the lowest. The results of entropy and standard deviation are similar for tools 4 and 8, indicating that they are the tools that provide the greatest stability in the cutting process. In addition, it can be noted that for many tools, entropy results were obtained in the lowest range of 0.3–0.4 and even below. This means that the cutting process is influenced not only by the microgeometry of the tool but also by the selection of the appropriate feed and depth of cut parameters. Therefore, the results obtained were statistically analysed.

Of all the cutting-force parameters, the mean value of force *MeanF* and the mean value of the upper envelope *MeanUpF* were selected for further analysis. The *MeanF* parameter determines the average force acting on the tool and the workpiece, while the *MeanUpF* parameter has a value close to the maximum cutting force values corresponding to the maximum thickness of the cut layer *h_max_*. Both these parameters are often used to analyse the cutting process.

First, the influence of the cutting-edge rounding radius *R* on the mean value and the mean value of the cutting-force maxima was analysed. Figure 8 shows collectively all the results as a function of the radius *R*. The dispersion of results for different values of *R* is large because the graphs include results for different margin widths and different cutting parameters. The curvature of the functions fitted to the results indicates the non-linear dependence of the cutting-force value on the radius *R*. Moreover, this dependence is nonmonotonic. For both the mean value of the force and the mean value of the force maxima, the effect of the radius *R* is similar. This means that, regardless of other geometric and technological parameters, the lowest cutting forces occur for the radius of the rounded cutting edge of about 18 micrometres. For values of *R that are* smaller and larger, the forces are slightly higher.

To more precisely analyse the influence of the radius *R* of cutting edge rounding on the cutting-force parameters, the results are presented for selected technological parameters and different values of *R* and *MAR*. Figure 9 compares the values of *MeanF* and *MeanUpF*. Analysing the presented results, it can be seen that the trend of change in the *MeanF* and *MeanUpF* values is the same in each case. Only the force values are different. In addition, it can be concluded from the presented results that the relationship of the cutting force with the rounding radius *R* is different for different values of the *MAR*. In contrast to the general trend shown in Figure 8, for large values of the margin width, the influence of radius *R* on the cutting force is reversed. A value of radius *R* of around 19 micrometres in most cases results in a minimisation of the cutting force. For the largest margin width and average radius of the rounded cutting-edge values, the force values are the largest. Additionally, for the largest *MAR* values, the influence of radius *R* on cutting force is not significant.

Analysis of the effect of the rounding radius of the cutting edge on the cutting force leads to conclusions similar to those obtained by other authors. In the work [16], the lowest cutting forces and the lowest tool wear were obtained in the steel milling process for the largest tested cutting-edge rounding radius of 15 micrometres. This confirmed that values of the rounding radius below this value are not beneficial for the cutting process. Additionally, in the work [17], it was confirmed that among the edge rounding radii of 4 to 15 micrometres studied, the least tool wear in steel milling occurred for an edge rounding radius of 12 micrometres. In turn, the work [20] showed that in the hard turning of bearing steel, the best cutting-process conditions were obtained for the middle value of the studied range of the radius of a cutting-edge rounding of 30 micrometres. In conclusion, other works show that for each tool and cutting process, the optimal value of the radius of the rounding of the cutting edge can be determined, and it is not the smallest value.

Next, the effect of the width of the margin on the cutting-force parameters was analysed. For this purpose, plots of the dependence of the mean force value and the mean value of the maxima on the width of the margin for different values of the radius *R* and technological parameters were made. The results are shown in Figure 10.

It can be concluded from the presented graphs that for both the mean value of the force and the mean value of the maxima, the influence of the margin width varies depending on the value of the radius *R*. For the smallest values of *R* close to 10 micrometres, the dependencies mostly have a parabola shape with a maximum for the margin width around 100 micrometres. The obtained dependencies are nonmonotonic and after exceeding *MAR* = 100 micrometres, the force slightly decreases. In contrast, for the value of the radius *R* about 19 micrometres, monotonic relationships close to linear were obtained for most technological parameters. This means that in this case, the cutting force increases proportionally to the increase in the width of the margin. For the largest values of radius *R*, the influence of the cutting force on the margin width is also different. For most of the technological parameters, dependencies were obtained in a non-linear form, usually monotonic. It can be seen that for margin widths below about 80 micrometres, the cutting force changes slightly. A stronger increase in the cutting force was observed only for margin values greater than 100 micrometres. This shows that for the largest cutting-edge rounding radii, the effect of margin widths of less than 80 micrometres is insignificant.

It can also be noted that similar relationships exist for both *MeanF* and *MeanUpF*. This means that the mean- and maximum-force values change very similarly. Technological parameters have a secondary influence and mainly affect force values and not the trend of change. Regardless of the technological parameters, in most cases, the obtained cutting-force variation curves have the same curvature. No research results on margin width were found. Only in paper [18] was the shape of the margin studied, but not the effect of its size on the cutting process. Therefore, the results of the study cannot be compared with the results of other works.

### 3.2. Cutting-Force Models and Response Surface

Due to the varying influence of the rounding radius *R* and the width of the margin *MAR* on the cutting-force parameters, a statistical analysis of the results was carried out. Mathematical models for *MeanF* and *MeanUpF* were developed as a function of technological parameters, *R* and *MAR* with the best fit, and a variance analysis was performed. For each cutting-force parameter, an equation was determined, and statistical parameters were determined to evaluate the resulting models. A significance level of α = 0.05 was adopted.

For the *MeanF* value of the cutting force, the following equation was obtained:*MeanF* = 81.5 − 18.61∙*R* + 2.450∙*MAR* + 1472∙*f_z_* + 83.4∙*a_e_* + 0.694∙*R*^2^ + 2.589 *MAR*∙*a_e_*,(6)
which fits very well with the experimental data, as the coefficient *R*^2^ = 0.9. A graphical representation of the model is shown in Figure 11 for the maximum and minimum values of the cutting parameters. Table 3 shows the results of the variance analysis of the received model. 

Studying the parameters of the analysis of variance, it can be seen that all the components of the equation are characterised by very small values of the *p*-value parameter. This means that they are statistically significant in the model. However, for only the quadratic component, a slightly higher value of the *p*-value parameter was obtained at an acceptable level. Analysis of the contribution and F-value parameters provides information on the significance of the influence of the parameters on the value of the function. The analysis shows that the margin value has the strongest influence on the mean force value. Its contribution in the model is more than 75%, and the F-value is more than three times higher than the other components of the equation. The second parameter, in terms of significance of influence on the average cutting force, is the technological parameter *a_e_*. Its contribution to the model is more than 10%. The contribution of the other components is at the level of 1–2%. It follows that the other parameters, such as feed per tooth and the rounding radius of the rounded cutting edge, do not have a significant effect on the cutting force. In addition, it can be seen that linear factors play a dominant role in the model. The sum of quadratic and two-way factors has a contribution of slightly more than 2%. However, analysing the graphical interpretation of the model for selected cutting parameters, one can see a significant curvature of the surface. It can be observed that as a function of *MAR*, the average force varies monotonically and, as a function of radius *R*, it has a nonmonotonic shape, which confirms the previous observations. The graphs in Figure 11 show that the *MeanF* function has a minimum for *R* ≈ 17 micrometres regardless of the other parameters. This means that, contrary to common trends, a tool for the machining of aluminium alloys should not have the smallest possible rounding radius of the cutting edge. Analysing the graphs in Figure 11, it can be seen that the surface in Figure 11a has more curvature than that in Figure 11b.

Statistical analysis was also performed on the average value of the maxima *MeanUpF* for which the following model was obtained:*MeanUpF* = 191 − 24.4∙*R* + 2.612∙*MAR* + 1358∙*f_z_* + 111.7∙*a_e_* + 0.915∙*R*^2^ + 3.186 *MAR*∙*a_e_*,(7)
which fits very well with the experimental data like *MeanF*, with coefficient *R*^2^ = 0.87. A graphical representation of the model is shown in Figure 12 for the maximum and minimum values of the cutting parameters, and Table 4 presents the results of the analysis of variance of the obtained model.

The graphs obtained from the *MeanUpF* function are very similar to those of the *MeanF* function, which confirms the previous observations. Both cutting-force parameters differ mainly in their values. The form of the model for both functions is the same. Linear factors dominate; in addition, there is one quadratic and one two-way factor. When comparing the graphs in Figure 11 and Figure 12, it can be seen that the response surfaces for the *MeanUpF* function have more curvature and are steeper. On the other hand, analysing Figure 11 and the results of the analysis of variance from Table 4, it can be deduced that the effect of *MAR* on *MeanUpF* is linear, and the effect of radius *R* is nonlinear and nonmonotonic. As with *MeanF*, for *MeanUpF*, the minimum of the function occurs for *R* ≈ 17 micrometres.

Analysing the statistical data in Table 4, it can be seen that the feed parameter *f_z_* has a high *p*-value. This means that, statistically, its effect on *MeanUpF* is much smaller than the other components of the equation. In addition, the radius *R* also has a slightly increased *p*-value parameter. The contribution and F-value analysis indicate that the margin *MAR* has a strong effect on the cutting force. The cutting parameter *a_e_* also has a strong influence, and this influence is stronger than on *MeanF*. The other components of the equation have a contribution of only 1–2%. In addition, the *MeanUpF* model has a slightly higher error than the *MeanF* model.

A comparative analysis of the real values (Figure 8, Figure 9 and Figure 10) and the obtained models (Figure 11 and Figure 12) proves that, in reality, the influence of tool microgeometry parameters varies more than the obtained models suggest, although the fitting error is only about 10%. Nevertheless, the analysis of both models leads to convergent conclusions regarding the influence of cutting-tool edge-geometry parameters and technological parameters on cutting-force parameters.

## 4. Conclusions

A comprehensive experimental study of the HSM milling process of 7075 aluminium alloy was carried out using end-mill cutters of different microgeometries. The radius of the cutting-edge rounding, and the width of the margin were modified. It was shown that both parameters of the edge geometry affect the cutting force, and that the effect varies. Cutting-force models were developed that can be used to design tools and select tools and cutting-process parameters for aluminium alloys. The obtained test results allowed us to formulate the following conclusions:The greater the width of the margin, the greater the dispersion of the cutting force, and the machining process is less predictable and stable (the standard deviation of force is the largest). This is because the greater the width of the margin, the greater the contact area between the tool and the workpiece material, and the greater the frictional force.For the largest cutting-edge rounding radii, a margin width of less than 80 micrometres has no effect on cutting force.The effect of margin width on cutting force is monotonic and can be approximated as linear: as the width of the margin increases, the cutting force increases.The influence of the rounding radius of the cutting edge is non-linear and non-monotonic with a minimum for *R* ≈ 17 micrometres.Using the smallest cutting-edge rounding radii does not lead to the smallest cutting forces.The lowest cutting-force values (mean and median maxima values) were for the smallest margin width and for the cutting-edge rounding radius *R* = 15–20 micrometres.

## Figures and Tables

**Figure 1 materials-16-03859-f001:**
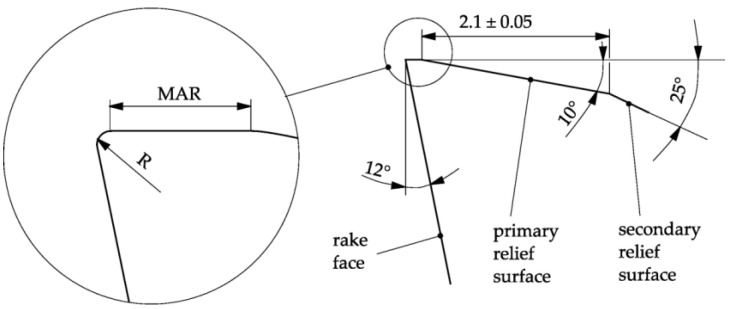
Selected dimensions of the cutter’s cross-section geometry, with designation of the variable values: *R*—radius of the rounded cutting edge and *MAR*—width of the cutter’s margin.

**Figure 2 materials-16-03859-f002:**
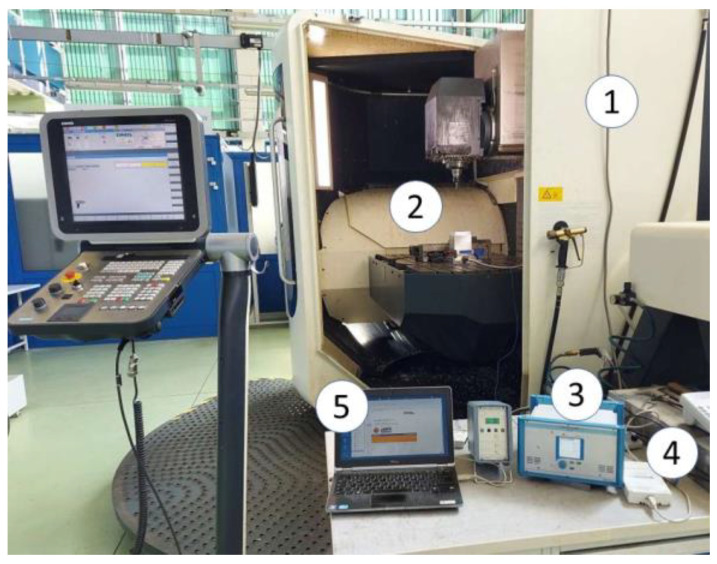
View of the test stand: (**1**) machine tool; (**2**) workspace with tool, dynamometer and sample; (**3**) charge amplifier; (**4**) A/C converter; and (**5**) computer.

**Figure 3 materials-16-03859-f003:**
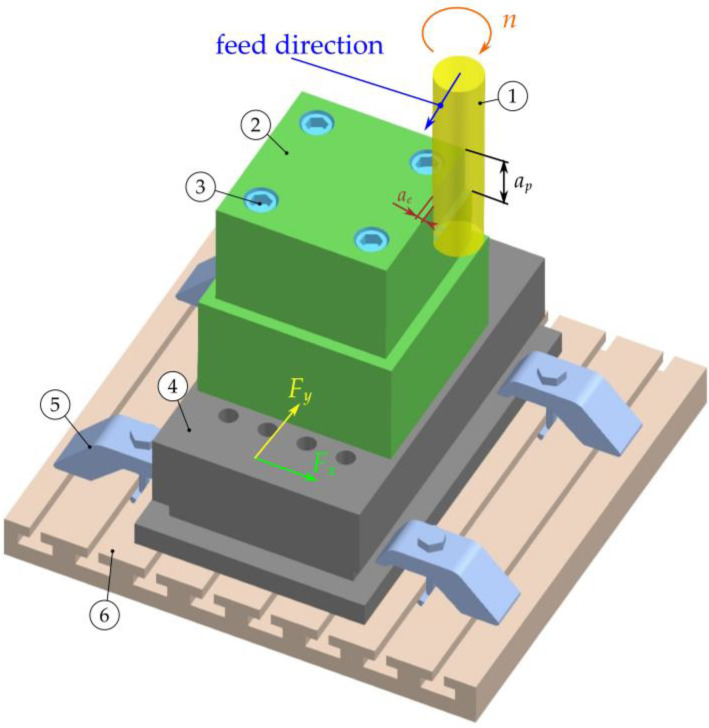
Diagram of the milling process: (**1**) tool; (**2**) workpiece; (**3**) clamping screws; (**4**) dynamometer; (**5**) clamping clamps; and (**6**) machine table.

**Figure 4 materials-16-03859-f004:**
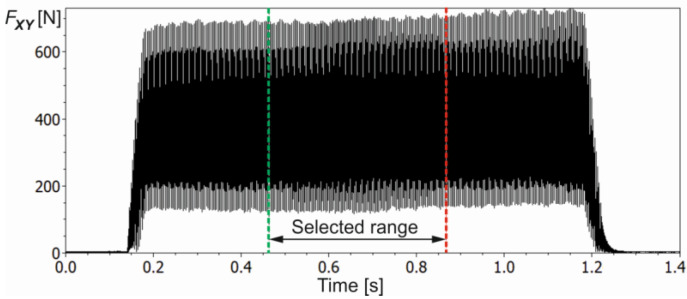
The course of the resultant cutting-force component *F_XY_* along with the selected time-series interval intended for analysis (without low-pass filter).

**Figure 5 materials-16-03859-f005:**
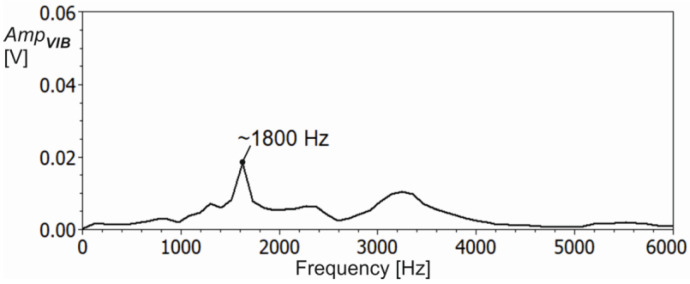
Spectral characteristics of the cutter-holder system after impulse modal test.

**Figure 6 materials-16-03859-f006:**
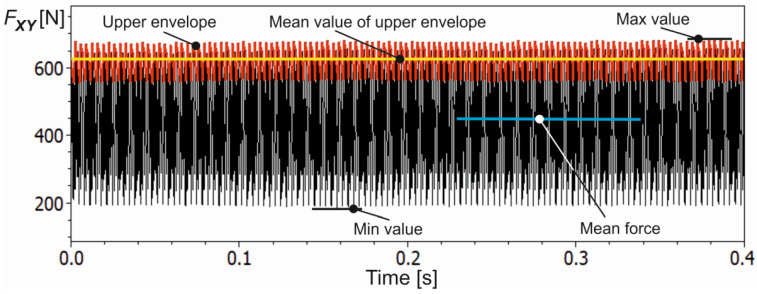
The course of selected range of the resultant cutting force *F_XY_* together with the mean line (yellow line) of the instantaneous force peaks (red line), maximum and minimum values of the force (signal is after low-pass filtering).

**Figure 7 materials-16-03859-f007:**
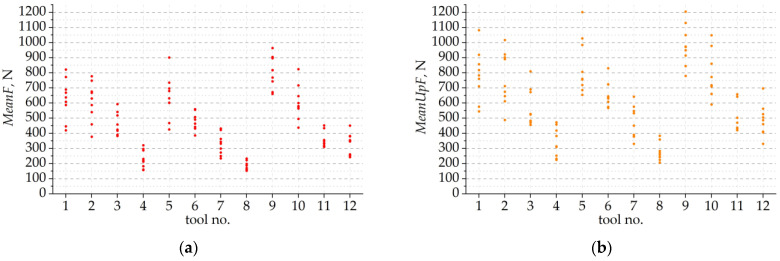
Values of the cutting-force parameters for the tested tools: (**a**) mean value; (**b**) mean value of the upper envelope; (**c**) maximum value; (**d**) minimum value; (**e**) dispersion; (**f**) standard deviation; (**g**) sample entropy; and (**h**) approximate entropy.

**Figure 8 materials-16-03859-f008:**
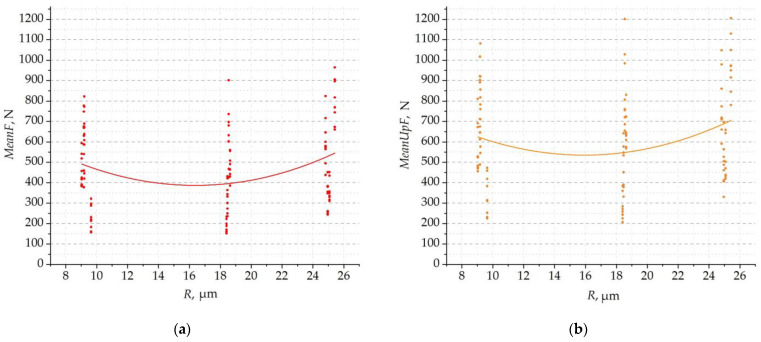
Cutting-force parameters in the function of radius *R*: (**a**) mean value of cutting force; and (**b**) mean value of the maxima.

**Figure 9 materials-16-03859-f009:**
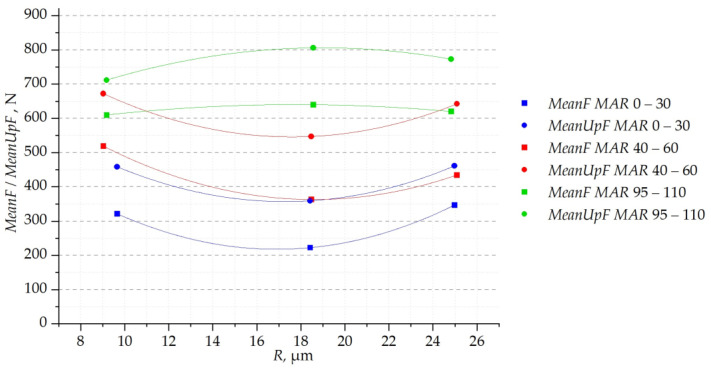
Relation between *MeanF* and *MeanUpF* and radius *R* for different margin ranges, feed *f_z_* = 0.1 mm/tooth and radial infeed *a_e_* = 0.7 mm.

**Figure 10 materials-16-03859-f010:**
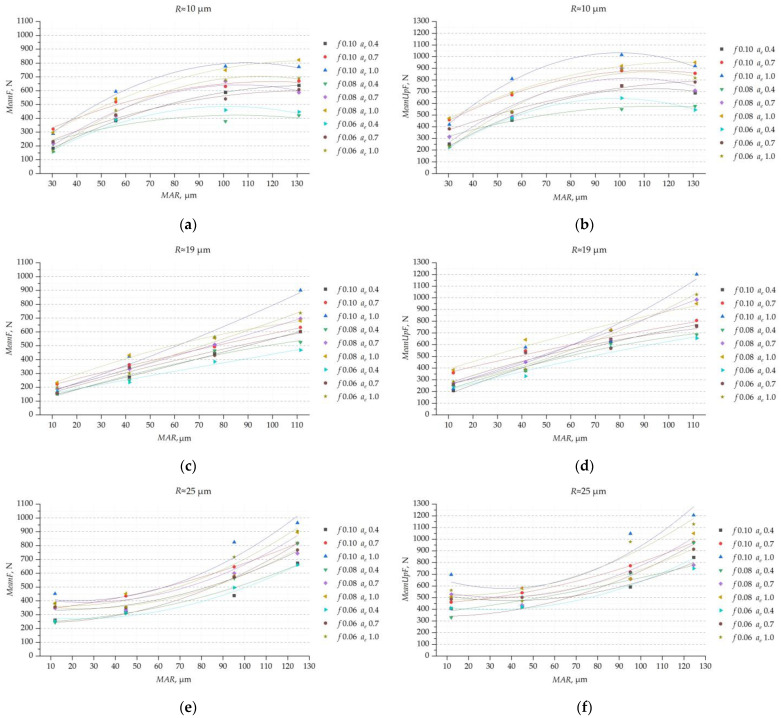
Relation between *MeanF* and *MeanUpF* and margin *MAR* for: (**a**) *MeanF* changes for *R* ≈ 10; (**b**) *MeanUpF* changes for *R* ≈ 10; (**c**) *MeanF* changes for *R* ≈ 19; (**d**) *MeanUpF* changes for *R* ≈ 19; (**e**) *MeanF* changes for *R* ≈ 25; and (**f**) *MeanUpF* changes for *R* ≈ 25.

**Figure 11 materials-16-03859-f011:**
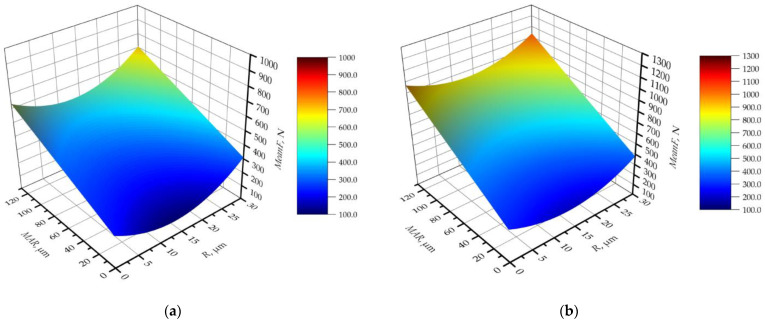
Mean value of cutting force in the function of *R* and *MAR* for: (**a**) the lowest cutting parameters, *f_z_* = 0.06 mm/tooth, *a_e_* = 0.4 mm; and (**b**) the highest cutting parameters, *f_z_* = 0.1 mm/tooth, *a_e_* = 1.0 mm.

**Figure 12 materials-16-03859-f012:**
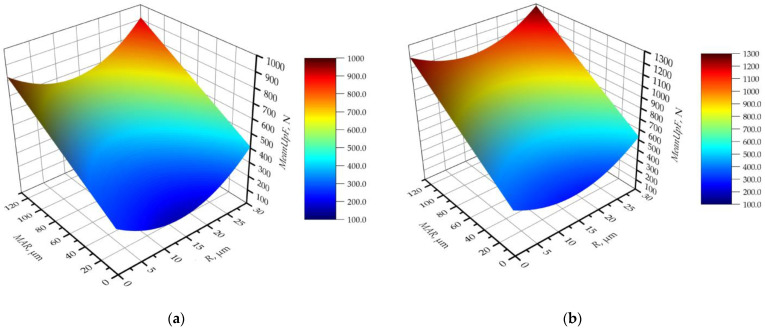
Mean value of maxima of cutting force in the function of *R* and *MAR* for: (**a**) the lowest cutting parameters, *f_z_* = 0.06 mm/tooth, *a_e_* = 0.4 mm; and (**b**) the highest cutting parameters, *f_z_* = 0.1 mm/tooth, *a_e_* = 1.0 mm.

**Table 1 materials-16-03859-t001:** Real values of the microgeometry parameters of the tool.

Tool No.	Real Value of the Cutting-Edge Radius *R*, μm	Real Value of the Margin Width *MAR*, μm
1	9.2	130.8
2	9.2	100.8
3	9.0	56.0
4	9.6	30.4
5	18.6	111.4
6	18.6	76.4
7	18.5	41.5
8	18.4	12.4
9	25.4	124.6
10	24.8	95.2
11	25.1	44.9
12	25.0	13.2

**Table 2 materials-16-03859-t002:** Experimental test plan.

Test Number	Feed per Tooth *f_z_*, mm/tooth	Radial Infeed *a_e_*, mm
1	0.08	0.7
2	0.10	0.4
3	0.06	0.7
4	0.08	1.0
5	0.08	0.7
6	0.06	0.4
7	0.08	0.7
8	0.10	1.0
9	0.06	1.0
10	0.08	0.7
11	0.08	0.7
12	0.08	0.4
13	0.10	0.7

**Table 3 materials-16-03859-t003:** Analysis of variance for the *MeanF* cutting-force model.

Term	DF	Seq SS	C (%)	Adj SS	Adj MS	F-Value	*p*-Value
Model	6	3,963,639	90.38%	3,963,639	660,607	158.12	0.000
Linear	4	3,851,565	87.82%	3,619,393	904,848	216.57	0.000
*R*	1	17,337	0.40%	125,175	125,175	29.96	0.000
*MAR*	1	3,319,034	75.68%	3,047,914	3,047,914	729.51	0.000
*f_z_*	1	62,446	1.42%	62,446	62,446	14.95	0.000
*a_e_*	1	452,747	10.32%	468,953	468,953	112.24	0.000
Square	1	40,969	0.93%	40,969	40,969	9.81	0.002
*R∙R*	1	40,969	0.93%	40,969	40,969	9.81	0.002
2-Way	1	71,105	1.62%	71,105	71,105	17.02	0.000
*MAR∙a_e_*	1	71,105	1.62%	71,105	71,105	17.02	0.000
Error	101	421,979	9.62%	421,979	4178		
Total	107	4,385,618	100.00%				

**Table 4 materials-16-03859-t004:** Analysis of variance for the *MeanF* cutting-force model.

Term	DF	Seq SS	C (%)	Adj SS	Adj MS	F-Value	*p*-Value
Model	6	5,320,906	87.19%	5,320,906	886,818	114.61	0.000
Linear	4	5,142,031	84.26%	4,829,506	1,207,376	156.03	0.000
*R*	1	49,224	0.81%	228,537	228,537	29.53	0.000
*MAR*	1	4,315,226	70.71%	3,934,427	3,934,427	508.46	0.000
*f_z_*	1	53,087	0.87%	53,087	53,087	6.86	0.010
*a_e_*	1	724,495	11.87%	749,672	749,672	96.88	0.000
Square	1	71,187	1.17%	71,187	71,187	9.20	0.003
*R∙R*	1	71,187	1.17%	71,187	71,187	9.20	0.003
2-Way	1	107,688	1.76%	107,688	107,688	13.92	0.000
*MAR∙a_e_*	1	107,688	1.76%	107,688	107,688	13.92	0.000
Error	101	781,524	12.81%	781,524	7738		
Total	107	6,102,430	100.00%				

## Data Availability

Data available on request.

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
