# Peer review of "Influence of Cutting-Edge Microgeometry on Cutting Forces in High-Speed Milling of 7075 Aluminum Alloy"

_materials, 2023, doi:10.3390/ma16103859_

Round 1
Reviewer 1 Report
1. The uncoated tools with sharp edge are usually used for the aluminium alloy, and the coated tools usually are unrecommended. The author claimed that the used tools were coated with a PCD coating after abrasive jet machining technology, so which coating types was used and why used the coating tools?
2. The measurement of cutting edge corner radius is a challenging job, from table 1, the real value of corner radius, some values are very close, such as 9.2 and 9.18, 18.42 and 18.47, so how to ensure the measurement accuracy of corner radius?
3. In the paper, the milling experiment is face milling of side milling operation? Pleasure provide a diagrammatic sketch or machining picture to illustrate.
4. The cutting force in X and Y direction should be illustrated in the paper, which is the tangential or radial. The dynamometer 9257B can record the force components in three direction, include the Z direction, so why the author ignores the cutting force in Z direction.
5. The cutting frequency of cutter flutes is 900Hz, the cutoff frequency of 1150Hz was applied for the force signal, the cutoff frequency is lower than three times of cutting frequency, how to ensure the force signal realism?
6. The meaning of abscissa in figure 6 should be mark in the figure.
No
Author Response
Thank you very much for your careful analysis of the manuscript and your valuable comments, which will improve the quality of the publication. All comments have been applied to the text of the manuscript. In addition, we send responses to the comments.
- The uncoated tools with sharp edge are usually used for the aluminium alloy, and the coated tools usually are unrecommended. The author claimed that the used tools were coated with a PCD coating after abrasive jet machining technology, so which coating types was used and why used the coating tools?
Response: In practice, uncoated cutters with polished chip groove surfaces are usually used for machining aluminum alloys although coated cutters with a thin coating dedicated to machining aluminum alloys are increasingly being offered. In the case of the milling cutters under study, it was not possible to use a stream polishing process because this process could negatively affect the width and shape of the margin and the size of the rounding radius. Hence, the decision was made to use a protective coating made by PVD technology. A ZrN coating with a hardness of 2800 HV and a thickness of only 2 micrometers was used. It is bio-compatible, extremely tough and highly resistant against abrasive wear. It is a coating dedicated to cutting non-ferrous materials. This reduced the adhesion phenomena in the milling process. In addition, the small thickness of the coating did not affect the microgeometry of the tools. The ZrN coating replaced the polishing process.
- The measurement of cutting edge corner radius is a challenging job, from table 1, the real value of corner radius, some values are very close, such as 9.2 and 9.18, 18.42 and 18.47, so how to ensure the measurement accuracy of corner radius?
Response: The radius of the rounded cutting edge was measured using Alicon's InfiniteFocus microscope. The results of the radius measurements were given by the microscope to the precision of hundredths of a micrometer. However, the standard deviation of repeated measurements for radius measurements using this microscope under the adopted measurement conditions was equal to 0.15 micrometer. Hence, the values in Table 1 were corrected and rounded to tenths of a micrometer.
- In the paper, the milling experiment is face milling of side milling operation? Pleasure provide a diagrammatic sketch or machining picture to illustrate.
Response: In the experiment, side milling with constant axial depth of cut was used. A schematic drawing was added to the manuscript showing the positioning of the cutter relative to the workpiece and the directions of the working movements.
- The cutting force in X and Y direction should be illustrated in the paper, which is the tangential or radial. The dynamometer 9257B can record the force components in three direction, include the Z direction, so why the author ignores the cutting force in Z direction.
Response: The dynamometer allows measurement of 3 components of the cutting force. The Z-axis component was also measured, but its value was constant for all tools, so it was skipped in the analyses. In addition, the Z component of the cutting force is a passive component. Only the X and Y components were analyzed because they act in a plane containing two velocity vectors; the feed rate and the cutting speed. The overall geometry of the blade was not changed so the distribution of the cutting force components was constant. In addition, the resultant force XY is the bending force of the tool and the force acting on the workpiece. Hence, only the resultant XY was analyzed, because in practice its value is used in calculations of tool stiffness and strength, and in calculations of workpiece deflection, especially of a thin-walled workpiece. A drawing showing the milling method was added to the manuscript and the X and Y directions of the measurement of the cutting force components were marked in this drawing.
- The cutting frequency of cutter flutes is 900Hz, the cutoff frequency of 1150Hz was applied for the force signal, the cutoff frequency is lower than three times of cutting frequency, how to ensure the force signal realism?
Response: The rotational speed of the cutter was 300 rot/s (18 000 rot/min). With three flutes of the cutter, this gives a frequency of material removal by individual flutes of 900 Hz. Therefore, the frequency of 900 Hz is the frequency of the blades and not the rotation frequency of the tool. Since the resonance frequency of the cutter-holder system was about 1800 Hz, we decided to use a cut-off frequency for the low-pass filter equal to 1150 Hz to remove the dynamic force components (tool vibrations) due to regenerative chatter.
- The meaning of abscissa in figure 6 should be mark in the figure
Response: Thank you for your comment. The labeling of the abscissa axis as a tool number has been added to Figure 6.
Reviewer 2 Report
In this paper the authors tried to analyze the influence of rounding radius of cutting edge, and the width of the margin width on the cutting force parameters. The experimental tests were carried out using central composite fractional design. The variable were radius of cutting edge, margin length, feed per tooth and radial in feed. The cutting speed and rotational speed were chosen as constants. In toal 156 tests were carried out for 12 different tools. The force components were measured in the X and Y directions. to characterise the cutting process, the mean force value MeanF and the mean value of the upper force envelope composed of the local force peaks value MeanUpF, corresponding to the successive cutter teeth, were calculated. In addition, the maximum MaxF, minimum MinF and dispersion DispF values of the resultant cutting force were determined. In addition, for each force course, the standard deviation StdDevF, sample entropy SampEntrF, and approximate entropy AppEntrF from the course of the resultant cutting force were determined. Moreover, Mathematical models for MeanF and MeanUpF were developed as a function of techno logical parameters, R and MAR with the best fit, and a variance analysis was performed. For each cutting force parameter, an equation was determined, and statistical parameters were determined to evaluate the resulting models. As a result, the authors stated that the greater the width of the margin, the greater the dispersion of the cutting force, and the machining process is less predictable and stable. The effect of margin on cutting force is monotonic, however this effect is non-linear and non-monotonic for the rounding radius of the cutting edge. As the width of the margin increases, the cutting force increases.
In my opinion the experiment performed in this paper by considering the proposed parameters can be helpfull to better understand the vibration phenomenon during milling operation. The methodolgy is clearly described. The conclusions consistent with the evidence and arguments presented and they address the main question posed. All the references all appropriate. There is no additional comments on the tables and figures.
The paper is clear, concise, and it is well-written. The introduction section is very well-written and it is based on the theory and the appropriate references are used and explained in detail for better underestading of readers. The materials and method section is also written clearly, the objectives clearly stated, applied methods are advanced. The results are sufficiently discussed and statistically analyzed. The conclusions well supported by the data presented
However, I have some comments:
1. In abstract the it is written "rounding radius of cutting edge". However, the authors use different terms for this. For example in Table 1 it is "Corner value". This is confusing for reader. Please check the paper and correct it and use the same term.
2. In abstract the it is written "width of the margin width". However, the authors use different terms for this. For example in Table 1 it is "margin length". This is confusing for reader. Please check the paper and correct it and use the same term.
3. Page 3, lines 141-143: "The microgeometry was made using abrasive jet machining technology and the tools were coated with a PVD coating in order to minimise friction." What was the material used for coating? It is calimed that they coated to minimize the friction. So, why the effect of coating on the friction and vibration is not reported in the paper?
4. Page 5, line 163-165. Why did the cutting speed and rotational speed were chosen to be constants? Why the v=900 m/min and n=18000 rpm were chosen? Why the effect of these papermeters was not investigated on the cutting forces? Although these parameters affect the cutting force.
5. Please write for Figure 3 and 5 capture, what were the cutting conditions.
6. Page 9, lines 282-285: "In contrast, the highest StdDevF values were obtained for tools No. 1, 5, 6 and 10, which are the tools with the largest margin values. Hence, it can be concluded that the larger the margin, the greater the dispersion of the cutting force, and the machining process is less predictable and stable." However, this statement contradict with the information demonstrated in Table 1. According to this statement the highest StdDevF values should be for tools No 1,2,5 and 9. And this is different from the authors' statement.
7. The results should be compared with the results of other researches related to their experiment.
Author Response
Thank you very much for your careful analysis of the manuscript and your valuable comments, which will improve the quality of the publication. All comments have been applied to the text of the manuscript. In addition, we send responses to the comments.
- In abstract the it is written "rounding radius of cutting edge". However, the authors use different terms for this. For example in Table 1 it is "Corner value". This is confusing for reader. Please check the paper and correct it and use the same term.
Response: Thank you for your comment. We have thoroughly revised the text of the manuscript and standardized the designation of the cutting edge rounding radius. We have made corrections to the text of the manuscript.
- In abstract the it is written "width of the margin width". However, the authors use different terms for this. For example in Table 1 it is "margin length". This is confusing for reader. Please check the paper and correct it and use the same term.
Response: Thank you for your attention. We have carefully revised the text of the manuscript and standardized the margin size labelling. We have adopted the term margin width.
- Page 3, lines 141-143: "The microgeometry was made using abrasive jet machining technology and the tools were coated with a PVD coating in order to minimise friction." What was the material used for coating? It is calimed that they coated to minimize the friction. So, why the effect of coating on the friction and vibration is not reported in the paper?
Response: In the case of tools designed for machining aluminum alloys, polishing or coating of the tool is usually used. This is done to minimize the phenomenon of adhesion between the workpiece material and the tool. In the case of the tested milling cutters, a protective coating dedicated to machining aluminum alloys was applied with the goal of minimizing the coefficient of friction. A coating was used to eliminate the effect of material adhesion on the test results. A coating dedicated to machining non-ferrous materials was used. It is a coating made by PVD technology, ZrN coating material, coating thickness of 2 micrometers. The coating of the cutter was made instead of the chip groove polishing process, which is most commonly used for this type of tool in machining Aluminum alloys. The coefficient of friction was not tested and the effect of the coating on friction was not studied. A protective coating was used because the blade could not be polished because the polishing process could change the width and shape of the margin.
- Page 5, line 163-165. Why did the cutting speed and rotational speed were chosen to be constants? Why the v=900 m/min and n=18000 rpm were chosen? Why the effect of these papermeters was not investigated on the cutting forces? Although these parameters affect the cutting force.
Response: Cutting speed is a technological parameter mainly related to the tool material and the workpiece material. Hence, the cutting speed value is determined for the cutting tool material - workpiece material pair. Therefore, the cutting speed of 900 m/min was established based on the tool manufacturer Engram's data as the recommended cutting speed for HSC cutting of aluminum alloy. In addition, the effect of cutting speed was not studied due to the large number of trials, which would have increased at least 3 times after taking into account the change in cutting speed. Investigating the effect of cutting speed on cutting force when milling aluminum alloy with tools of different microgeometries will be the next stage of research work.
- Please write for Figure 3 and 5 capture, what were the cutting conditions.
Response: Figures 3 and 5 (now 4 and 6) were made for the following cutting conditions: edge rounding radius 25 micrometers, margin width 45 micrometers, feed per tooth 0.1 mm/tooth, depth of cut 0.7 mm. These data were included in the manusctipt.
- Page 9, lines 282-285: "In contrast, the highest StdDevF values were obtained for tools No. 1, 5, 6 and 10, which are the tools with the largest margin values. Hence, it can be concluded that the larger the margin, the greater the dispersion of the cutting force, and the machining process is less predictable and stable." However, this statement contradict with the information demonstrated in Table 1. According to this statement the highest StdDevF values should be for tools No 1,2,5 and 9. And this is different from the authors' statement.
Response: Thank you for the accurate comment. Truly, our interpretation of the results was not precise. Therefore, we have corrected the description of the test results in terms of the analysis of the standard deviation parameter StdDevF. The largest values of standard deviation were recorded for tools 1, 2, 5, 6, 9 and 10. These are the tools with the largest margin width values above 76 micrometers. Hence, it can be concluded that the greater the margin width, the greater the dispersion of the cutting force, and the machining process is less predictable and stable.
- The results should be compared with the results of other researches related to their experiment.
Response: As suggested by the reviewer, the results of the study were compared with those obtained by other researchers. Consequently, the description of the study results in the manuscript was supplemented. No research results were found with regard to margin width. In paper [18], the shape of margin was studied but the effect of its size on the cutting process was not investigated. Therefore, it is not possible to compare the research results with those of other works. Analysis of the effect of the rounding radius of the cutting edge on the cutting force leads to conclusions similar to those obtained by other authors. In the work [16], the lowest cutting forces and the lowest tool wear were obtained in the steel milling process for the largest tested cutting edge rounding radius of 15 micrometers. This confirmed that values of the rounding radius below this value are not beneficial for the cutting process. Also in the work [17], it was confirmed that among the edge rounding radii from 4 to 15 micrometers studied, the least tool wear in steel milling occurred for an edge rounding radius of 12 micrometers. In turn, the work [20] shows that in hard turning of bearing steel, the best cutting process conditions were obtained for the middle value from the studied range of the cutting edge rounding radius of 30 micrometers. In conclusion, other works show that for each tool and cutting process, the optimal value of the cutting edge rounding radius can be determined, and it is not the smallest value.
Round 2
Reviewer 1 Report
all comments have been revised.
No
Reviewer 2 Report
The authors have clearly answered to the questions. In my opninion the paper is suitable for publication as it is.